# Challenging the Biomimetic Promise—Do Laypersons Perceive Biomimetic Buildings as More Sustainable and More Acceptable?

**DOI:** 10.3390/biomimetics10020086

**Published:** 2025-02-01

**Authors:** Michael Gorki, Olga Speck, Martin Möller, Julius Fenn, Louisa Estadieu, Achim Menges, Mareike Schiller, Thomas Speck, Andrea Kiesel

**Affiliations:** 1Cluster of Excellence *liv*MatS @ FIT—Freiburg Center for Interactive Materials and Bioinspired Technologies, University of Freiburg, Georges-Köhler-Allee 105, 79110 Freiburg, Germany; olga.speck@biologie.uni-freiburg.de (O.S.); martin.moeller@livmats.uni-freiburg.de (M.M.); julius.fenn@psychologie.uni-freiburg.de (J.F.); louisa.estadieu@livmats.uni-freiburg.de (L.E.); thomas.speck@biologie.uni-freiburg.de (T.S.); kiesel@psychologie.uni-freiburg.de (A.K.); 2Institute for Psychology, University of Freiburg, 79085 Freiburg im Breisgau, Germany; 3Institute for Philosophy, University of Freiburg, 79085 Freiburg im Breisgau, Germany; 4Plant Biomechanics Group @ Botanic Garden Freiburg, University of Freiburg, Schänzlestr. 1, 79104 Freiburg im Breisgau, Germany; 5Öko-Institut, Institute for Applied Ecology, Merzhauser Str. 173, 79100 Freiburg, Germany; 6Institute for Computational Design and Construction, University of Stuttgart, Keplerstr. 11, 70174 Stuttgart, Germany; 7Cluster of Excellence IntCDC, University of Stuttgart and Max Planck Institute for Intelligent Systems, Keplerstr. 11, 70174 Stuttgart, Germany

**Keywords:** biomimetic bias, biomimetic promise, layperson perception, natural-is-better bias

## Abstract

This study investigates whether or not laypersons perceive biomimetic buildings as more sustainable and acceptable, a notion termed the “biomimetic promise”. Employing an experimental design (*N* = 238), we examined assessments of three real-world biomimetic buildings at the University of Freiburg, namely the Fiber Pavilion in the Botanic Garden, the ceiling of the former zoology auditorium, and the Biomimetic Shell at the technical faculty. Participants were divided into two groups: one group was informed about the biomimetic nature of the buildings and the other group was not. Results showed no significant difference in perceived sustainability or acceptability between the two groups, favoring the hypothesis that there exists *no* “biomimetic bias”. Notably, with the exception of perceived sustainability comparing the pavilion and the auditorium, significant differences in assessments regarding sustainability and acceptability were observed between the buildings, emphasizing the importance of domain-specific factors in public judgments. These findings suggest that merely framing a technology as biomimetic does not inherently enhance its perceived sustainability or acceptability by laypersons. Instead, the study highlights the need for transparency and clear communication regarding sustainability benefits to gain societal acceptance of biomimetic technologies.

## 1. Introduction

In the current epoch of the Anthropocene, we face global challenges, especially related to sustainability. To address these challenges, there is a trend towards a return to nature [1]. Thereby, nature-derived solutions in terms of bioinspired and biomimetic approaches to solving technical problems were suggested [2]. In contrast to conventional engineered products, the inspiratory flow of ideas and knowledge from biological models to technical applications seems to promise more sustainable solutions [3,4,5]. However, as we will explain in detail below, this so-called biomimetic promise does not, per se, hold true and thus might be considered a “fallacy” when driving the assessment of technical solutions. In analogy to this fallacy and to the natural-is-better bias [6], we empirically test whether or not the assessment of laypersons is generally biased in the way that they consider bioinspired or biomimetic technical solutions as more sustainable and acceptable. An experimental study on the assessment of buildings as use cases shows that laypersons do not reveal a “biomimetic bias”. We conclude that framing a technical solution as biomimetic is not sufficient to warrant its public acceptability. Instead, biomimetic design does not only require considering sustainability, but also communication to the public needs to inform about sustainability (in terms of ecological, economic, and social benefits) explicitly. In the discussion section, we further elaborate on the implications for public communication regarding the sustainability of biomimetic technical solutions, relevant particularly to practitioners, such as science communicators.

### 1.1. The Biomimetic Promise

According to an influential paper by von Gleich et al. [4], there are three types of learning from nature: (i) “learning from the results of evolution” (e.g., hoop-and-loop fastener), (ii) “learning from the process of evolution” (e.g., the optimization method Evolution Strategy), and (iii) “learning from the success principles of evolution” (e.g., closed loop economy). As has been pointed out by several authors [4,7,8,9], learning from nature in general and from biological models in particular offers several potential benefits for novel technical solutions, claiming that biomimetic or bioinspired novel technologies and materials are innovative, transformative, aesthetically striking, low-risk, and highly sustainable. Here we focus on the last aspect, which is the core of the so-called *biomimetic promise*.

Before addressing the biomimetic promise in detail, we would like to clarify that novel technical solutions can be inspired from living nature in different ways, and therefore various terms are used in the literature. In this article, bioinspiration is understood as the transfer of an idea from a biological model to a technical application, biomimetics is based on the transfer of a functional principle to a biomimetic product, and biomimicry is defined as the transfer of strategies to sustainable design [7,10,11]. We are aware that the biomimetic promise was initially limited to biomimetic applications [4] but is increasingly attributed to bioinspired applications as well (e.g., [12]). For ease of reading, we will use the term biomimetics and biomimetic promise throughout the following text (even when addressing bioinspired technical solutions).

In recent years, there have been fundamental considerations and substantial discussions among experts in the field of biomimetics as to whether or to what extent the flow of knowledge from biology can lead to more sustainable technologies and products [7,8,9,12,13]. While some authors advocate that the biomimetic design process requires including sustainable assessments (ideally in terms of circular economy), other authors seem to suggest that biomimetic solutions might “inherit” sustainability [7,8].

Importantly, reasoning behind the biomimetic promise extends beyond functional heuristics, encompassing epistemic, normative, and emotional aspects [4,7,14]. According to Benyus [11], epistemic, normative, and emotional aspects become the most obvious where nature is considered not only as a model but also where nature is seen as a mentor and even a measure. From a philosophical point of view presented by Gerola et al. [14], “there are two general roles that nature can play in biomimetic design. The first is as a technical model that provides insight into functional principles that can be abstracted and translated into technology. The second is as a normative source of ecological principles that enable us to evaluate the ecological appropriateness of our technologies” [14] (p. 65). To conclude, the biomimetic promise states that technical solutions derived from nature would be more sustainable than conventional technical solutions.

### 1.2. In Danger of a Naturalistic Fallacy

Despite the increasing impact (and success) of biomimetics, the biomimetic promise remains theoretically contested, especially when it is seen as a normative source for ecological principles. Unsurprisingly, invoking living nature to judge technology appropriateness has been criticized for committing the naturalistic fallacy. Such inferences confuse what *is* with what *ought to be*, thereby projecting human teleological and anthropocentric interpretations onto nature and obfuscating normative judgments by inappropriately “naturalizing” them [7,15,16]. For example, Höfele [17] states that natural processes are not to be understood as sustainable per se, nor can they be attributed a normativity if one does not want to succumb to a naturalistic fallacy. Instead, Höfele derives a philosophy of biomimetics with reference to Kant’s concept of a technology of nature, which assumes that “the results of biological evolution are interpreted purposefully; only then can they be used as a model for purposefully designed technologies” [17] (p. 102).

Moreover, as has been frequently pointed out, biomimetic and bioinspired technical solutions are not, per se, sustainable and it is crucial to evaluate the actual sustainability of a technology independently from its biomimetic approach [2,4,17,18]. Thus, inferring a technology’s appropriateness solely based on its biomimetic nature or bioinspiration constitutes a fallacy. With a biological concept generator or model, a contribution to economic, ecological, and social sustainability cannot be a by-catch.

We therefore conclude that there are two main ways of conceptualizing the relationship between the biomimetic promise and sustainability: One way in which sustainability is taken as a mission statement when learning from living nature and another way in which sustainability is taken as an *inherent property* of biomimetic technologies. For the reasons pointed out above, we consider the latter as a *fallacy*.

### 1.3. A Biomimetic Bias?

In the current study, we explore whether or not laypersons might fall for such a fallacy. In psychology, fallacies are often studied as *biases*, that is, as heuristics that lead to misleading judgments and can thus affect attitudes [19,20]. In the following, we use the term *biomimetic bias*—such a bias would indicate that laypersons perceive a technical solution as more sustainable when merely framed as biomimetic.

#### 1.3.1. “Laypersons”

Perspectives: We consider it important to study a potential *biomimetic bias*, i.e., “laypersons” perception of biomimetic technologies, for at least two reasons: First, incorporating societal perspectives is generally vital in the context of sustainability, as ignoring laypersons’ assessments could undermine the legitimacy of innovation and thus eventually the social dimension of sustainability itself [21]. Second, understanding laypersons’ assessments is vital for the acceptance and market potential of biomimetic technical solutions, as expectations can only be met if they are properly understood. Despite their importance, the laypersons’ perspectives remain understudied [22]. Especially, as Speck et al. highlight, “The emotional perception is usually a crucial aspect for the market potential of innovative products. A positive connotation of biomimetic products might enable, or at least enhance, their market introduction” [7] (p. 11).

Consequently, it is not only important whether or not experts accept the premises of the biomimetic promise but also whether “laypersons” evaluate these technologies favorably along the lines of the biomimetic promise. Currently, to our best knowledge, there is no research on whether or not there is a biomimetic bias for laypersons, but in analogy to the well-studied *natural-is-better bias* it is plausible to expect such an effect. Due to the importance of emotional aspects, it is also sensible to assume that the strength of this effect might interact with other features of technical solutions, like, e.g., their aesthetic qualities or their context of use.

#### 1.3.2. The Analogy to the Natural-Is-Better Bias

There is a considerable amount of psychological research, especially in the field of risk perception, providing evidence for a so-called natural-is-better bias. Consumers tend to prefer products that are labeled and/or perceived as more natural in comparison to functionally similar products. However, this research has mainly focused on the assessment of specific products, that is, on products that are applied directly to the body (e.g., food, drugs, or cosmetics) and/or on products or environments, where the sensory characteristics of the products or environments perceived as more natural were considerably different from those perceived as less natural [6,23,24,25,26,27]. Furthermore, there is some evidence for the natural-is-better bias beyond consumer goods. Regarding injuries, there seems to be a bias towards naturally acquired injuries rather than non-natural ones, as there is, e.g., some evidence that people prefer to be injured by lightning in a hypothetical situation rather than getting the same injury through a downed power line [6,28].

Please note that the natural-is-better bias does not automatically imply a biomimetic bias. As described above, bioinspiration, biomimetics, and biomimicry focus on the transfer of ideas, functional principles, and strategies from living nature to technical solutions, while much of the research on the natural-is-better bias is focused on products that are merely based on “natural” or less processed ingredients or natural phenomena like, e.g., lightening [6,28]. Thus, while there is a large body of research on the natural-is-better bias [6,26], it remains unclear whether or not the natural-is-better bias generalizes to contexts where technical solutions are not directly applied to the human body and where “nature” does not primarily come into play as a direct source of material ingredients but as a concept generator.

#### 1.3.3. “Laypersons” Attitudes Towards Biomimetic Buildings

In the current study, we aim to explore whether or not there exists a biomimetic bias in laypersons’ assessment. In detail, we investigate whether or not biomimetic technical solutions are perceived as more sustainable and rated as more acceptable by laypersons. As a use case, we have chosen three real-world biomimetic buildings at the University of Freiburg, namely the Fiber Pavilion in the Botanic Garden, the ceiling of the former zoology auditorium, and the Biomimetic Shell at the technical faculty. Biomimetic buildings are receiving increased attention lately with the hope of supporting a shift into sustainable practices [29,30,31]. This trend is even more important, as the building sector is crucial, not only for tackling key sustainability challenges like climate change but also for creating environments that are economically and socially sustainable [32,33,34,35]. In summary, our use case of biomimetic buildings differs from the existing literature on the natural-is-better bias on two dimensions: (i) the use of biomimetic principles instead of “natural” materials, and (ii) the domain of application further apart from the human body.

#### 1.3.4. The Interplay of Perceived Sustainability and Acceptability

Furthermore, we use the data of the current study to assess whether or not the ratings of perceived sustainability and acceptability are positively correlated. The framework proposed by Huijts et al. [36] in the context of renewable energies provides evidence for the assumption that the perception of a technology as ecologically, economically, and socially sustainable is positively related to its acceptability. Ecological, economic, and social sustainability are addressed within this framework as parts of the perceived risks, costs, and benefits related to a technology. There is similar evidence from other areas like, e.g., smart grid technologies [37], climate engineering technologies [38], and information systems [39] for a strong relationship between the perceived sustainability of a technology and its acceptability and/or use. Consequently, we aim to test whether or not the same holds true when considering perceived sustainability and acceptability for (biomimetic) buildings.

### 1.4. Hypotheses

Based on these considerations, we preregistered three main hypotheses for our study (the preregistration can be found at OSF: https://doi.org/10.17605/OSF.IO/F2ZPD, registered on 31 May 2024).

**Biomimetic hypothesis:** A building that is framed as biomimetic will be perceived (a) as more sustainable and (b) as more acceptable.

**Interaction hypothesis:** The size of the biomimetic bias differs for the three types of building.

**Mediation hypothesis:** The increased perceived sustainability mediates the increased perceived acceptability. Foreshadowing the results, we did not test this hypothesis, because we did not observe the main effect of the biomimetic information on perceived sustainability and acceptability (as stated in the biomimetic hypothesis).

## 2. Methods

We conducted an experimental online study in spring of 2024, prepared by a pretest (*N* = 30) of all stimulus material and questionnaires in late 2023. The main results of the pretest are reported in Appendix B.

### 2.1. Power Analysis

We used the software G*power to determine the sample size. Based on a power calculation with a power (ß) of .95 and an α level of .05 for an ANOVA with repeated measures focusing on the between-subject factors, the optimal sample size required was 230 participants, assuming a small to medium effect size (f = .175) [40]. We estimated the effect size based on the pretest and existing literature on the natural-is-better bias. The pretest indicated a small to medium effect, while previous studies on the natural-is-better bias have reported effect sizes ranging from medium to large [41,42]. To account for potential drop-out in the case where participants might have prior knowledge of one of the presented buildings, we set the initial sample size to 250 participants.

### 2.2. Participants

We recruited a sample of 250 participants living in Germany online via the marketplace Prolific. We used the option, provided by Prolific, to collect a sample based on a gender quota, in order to be approximately representative in that regard as the option to distribute the study based on census data for even better representativity is not available on Prolific for German samples. All participants gave informed consent and received a payment of GBP 3 for an approximate overall study time of 20 min (15 min for the first part, 5 min for the follow-up) after completing the online survey. The requirements for participating in the study were a fluent level of proficiency in German. Individual data sets were excluded if participants indicated that they knew one of the buildings specified in the study beforehand or if the time needed to complete the study seemed unusually high. According to these criteria, 12 participants were excluded, because they indicated previous knowledge of at least one of the buildings. Thus, the final sample consisted of 238 participants (M_*age*_ = 31.00, SD_*age*_ = 9.65; for gender distribution, see Figure 1). Furthermore, as reported in Appendix C, Appendix D and Appendix E, our sample had a substantial degree of diversity in terms of ethnicity, income, and socio-political attitudes.

### 2.3. General Procedure

Participants completed an online study consisting of the main study and a follow-up.

#### 2.3.1. Main Study

In the main study, participants first gave informed consent (Figure 2). Then, participants were presented with information on three biomimetic buildings and rated each building. Participants were randomly divided into two groups of equal size. Both groups received exactly the same information regarding each building, except for the framing as biomimetic. In detail, one group was additionally informed that the buildings were inspired by nature and derived from a specific biological model. (Please note that we did not use the specialist term “biomimetic” so that layperson could understand the framing). Both groups received the following information for each building: (a) a photograph of the building (cf. Figure 3, Figure 4 and Figure 5), (b) general information about the building’s construction and purpose, and (c) information about advantages and limitations regarding the sustainability of the building. We controlled for potential order effects by varying the order of buildings in both groups and randomly assigning each participant to one of the six possible orders in the presentation of the three buildings. Thus, the only difference in how both groups were informed about the buildings was the experimental manipulation of framing the buildings as biomimetic only in one group.

After receiving the information for each building, participants rated the building regarding acceptability and perceived sustainability by answering two self-developed questionnaires. Furthermore, as a manipulation check, participants were asked what they assumed the buildings were inspired by. After completing the scales for each building, participants provided information on demographic data, responded to questions regarding their socio-political attitudes, and answered the question of how important sustainability is to them. Finally, participants were asked to indicate any potential technical problems or further comments and were reminded that the study included a follow-up part.

#### 2.3.2. Follow-Up

To test the stability of participants’ assessment regarding acceptability and perceived sustainability, we conducted a follow-up measurement. The follow-up started 10 days after the main study, and participants had three days for completion. In the follow-up, participants were reminded of the three buildings by again being presented the name and a photograph of the building. Then, participants again rated the acceptability and perceived sustainability.

### 2.4. Stimulus Material

To maximize ecological validity, we used photographs and textual information about three real-world biomimetic buildings at the University of Freiburg, Germany. We took care to present factually sound information on the actual buildings while wording the textual information. For each building, we structured the textual information into three parts: (a) general information about the construction (for all participants), (b) for only half of the participants, information that the building is biomimetic, and (c) information on the sustainability of the building (for all participants). In the following, we depict the respective information for each building (translated to English).

#### 2.4.1. Fiber Pavilion

**General information about the construction**. This building is a pavilion located in a Botanic Garden in Germany. The pavilion is intended for various types of event. It covers an area of 46 square meters and weighs 1.5 tons. Particular emphasis was placed on the lightweight nature of the construction elements. The ceiling load is supported by 15 fiber elements. These fiber elements are made from fibers of various plants, reinforced with synthetic resin. The fiber elements were designed on a computer and wound by robots. The fiber elements support a ceiling element made of transparent plastic, which protects against rain.

**Inspired by nature (visible for biomimetic group only)**. The pavilion is inspired by nature, meaning it is derived from biology: The structure of the wound fibers was modeled after the wood structure of certain cacti.

**Information on Sustainability**. The plant fibers used are made from 100% renewable raw materials and come from regional cultivation. Additionally, the plant fibers are biodegradable. The resin used is a petroleum-based product. Furthermore, the resin is neither biodegradable nor recyclable. The transparent ceiling elements are made from a durable plastic that can be recycled by melting it down. However, recycling the plastic from which the ceiling elements are made requires a significant amount of energy. The robot-assisted manufacturing is cost-effective but requires specialized knowledge that cannot be provided by craftsmen and expensive machinery.

#### 2.4.2. Ceiling of an Auditorium

**General Information about the construction**. This building is an auditorium at a German university. The lecture hall is intended for lectures. It is circular, has a floor area of 440 square meters, and accommodates up to 383 people. The ceiling construction is made of reinforced concrete. Particular emphasis was placed on reducing the weight of the concrete ceiling. The ceiling has thicker “ribs” only in the areas where special forces act on it, while there are recesses in the unloaded areas. The ceiling was made using a concrete formwork.

**Inspired by nature (visible for biomimetic group only)**. The auditorium ceiling is inspired by nature, meaning it is derived from biology: The structure of bones served as a model for the structure of the auditorium ceiling.

**Information on sustainability**. It is a concrete construction. The emission of climate-damaging greenhouse gases during the construction of the auditorium ceiling was about half that of an ordinary prestressed concrete or hollow-core slab of the same size. The total energy expenditure during the construction of the auditorium ceiling was about three-quarters compared to a comparable prestressed concrete or hollow-core slab. Due to the complex ceiling structure, a large amount of wood was needed for the formwork. This significantly contributes to the fact that the land use (mainly forestry land) for the auditorium ceiling is about 40 times higher than for a comparable prestressed concrete or hollow-core slab. The pure material costs of the auditorium ceiling do not differ significantly from those of a comparable prestressed concrete or hollow-core slab. The labor costs for the construction of the auditorium ceiling are three times higher than for the construction of a comparable prestressed concrete or hollow-core slab.

#### 2.4.3. Wooden Shell

**General information about the construction**. This building is a wooden pavilion located in the outdoor area of a German university. The pavilion serves as a year-round event venue. The shell-shaped pavilion covers an area of 200 square meters and spans 16 m without additional supports. The wooden shell of the pavilion consists of individual panels and weighs only 27 kg per square meter of shell surface while being very stable. Particular emphasis was placed on the lightweight nature of the construction elements and good acoustics. The components were designed on a computer and prefabricated using robots.

**Inspired by nature (visible for biomimetic group only)**. The wooden pavilion is inspired by nature, meaning it is derived from biology: the model for the modular design and shape was the skeleton of sea urchins.

**Information on sustainability**. The wood material used for the shell replaces materials such as steel and concrete, which produce large amounts of climate-damaging CO_2_. Due to the special lightweight structure of the wooden shell, material consumption is reduced by more than 50% compared to an ordinary wooden construction. The concrete used for the floor slab partially consists of recycled old concrete. The wood material weathers and changes its appearance when exposed to sunlight and weather conditions. For example, it can turn gray over time. The floor slab is still made of concrete, which only partially consists of recycled material and causes the emission of climate-damaging CO_2_. The entire construction is designed to be easily disassembled and reusable. The robot-assisted manufacturing requires specialized knowledge that cannot be provided by craftsmen and expensive machinery.

### 2.5. Measurements

#### 2.5.1. Sustainability Measurement

As there was no established applicable questionnaire available to assess the perceived sustainability of each of the three buildings, we used a self-constructed sustainability scale. The scale included four items that assessed the general perceived sustainability and the perceived ecological, economic, and social sustainability, respectively. The items were answered on a seven-point scale ranging from one “not sustainable at all” to seven “very strongly sustainable”. The scale score was obtained by calculating the mean of all items. We constructed the scale based on the three-pillar model of the sustainability [43]. In the pre-test, the scale had an internal consistency of Cronbach’s α > .7 for all buildings, indicating acceptable internal consistency. In the main study, Cronbach’s α was > .8 for all three buildings.

#### 2.5.2. Acceptability Measurement

As there was no established applicable questionnaire available to assess the acceptability of each of the three buildings, we used a self-constructed acceptability scale. The scale consisted of three items that were conceptually based on a classification of different types of energy technology acceptance [44]. These three items referred to the three levels of general acceptance, municipal acceptance, and state acceptance. An example item of the scale is “To what extent would you be in favor of or against the state promoting the construction of buildings of the type described?” The items of the scale were answered on a seven-point scale, with the response alternatives ranging from “strongly against” to “strongly in favor”. The scale score was obtained by calculating the mean of the three items. In the pre-test of the study, the scale achieved an acceptable internal consistency of Cronbach’s α > 0.7 for all three buildings. In the main study, Cronbach’s α was >0.9 for all three buildings.

#### 2.5.3. Measurements of Socio-Political Attitudes

We assessed a number of socio-political measure to (a) describe the participant sample in more detail and (b) to make sure that the two experimental groups did not differ between possible confounding variables for sustainability-related attitudes and assessments (e.g., [45,46,47]) (see Appendix D). The analyses regarding potential differences between the two groups did not indicate any group differences, except for self-rated own health (see Appendix E). Thus, socio-political attitudes are very unlikely to have played any substantial role as confounders.

#### 2.5.4. Statistical Analyses

We preregistered our hypotheses and planned statistical analyses on OSF (https://doi.org/10.17605/OSF.IO/F2ZPD, registered on 31 May 2024). According to the preregistration, we followed a multiverse analysis approach [48] and used both frequentist and Bayesian analyses. We specifically used Bayesian analyses to be able to also provide evidence *in favor* of the null hypothesis. Notably, this approach allows us to distinguish between absence of evidence and evidence of absence in case of non-significant frequentist results (e.g., [49,50]) (see also Appendix F). We consider 0.05 as the critical α level for all frequentist inferential statistics. For all Bayesian statistics, we consider Bayes factors below 3 for either H0 or H1 as inconclusive and above 10 as strong evidence for the respective hypothesis [51]. We performed all statistical analyses for testing our hypotheses in JASP 0.18.1 [52], while we performed the preprocessing of data and further analyses on socio-political measures in R 4.3.2 [53].

## 3. Results

As preregistered, we conducted 2 X 3 mixed measures ANOVAs (rmANOVA) with the between subject factor information that the buildings are bioinspired (yes/no) and the within subject factor type of building, for both perceived acceptability (Figure 6) and for perceived sustainability (Figure 7), as dependent variables. Non-significant Mauchly tests confirmed sphericity for both perceived acceptability and perceived sustainability, while non-significant Levene’s tests confirmed the homogeneity of error variances. For Bayesian rmANOVAS, we used JASPs default uniform prior, with the default r scale coefficient priors of 0.5 for fixed effects, 1 for random effects, and 0.354 for covariates.

### 3.1. Differences in Perception Between Buildings

The rmANOVAs revealed that there was a main effect of the repeated measures factor type of buildings, that is, participants rated the acceptability of the buildings, *F*(1, 236) = 57.21, *p* < .001, $\eta_{p}^{2}$ = .195, as well as the sustainability of the buildings, *F*(1, 236) = 55.59, *p* < .001, $\eta_{p}^{2}$ = .191, significantly different between the buildings. Bayesian rmANOVAs furthermore revealed strong evidence in favor of a model with the repeated measures factor of the buildings compared to the null model, that is, for assuming that there are differences regarding both the perceived acceptability, *BF*_01_ > 100, and the perceived sustainability, *BF*_01_ > 100, between the different buildings. Figure 6 and Figure 7, confirmed by post hoc tests, indicate that the Biomimetic Shell was rated as more sustainable, *p*_*bonf*_ < .001, *BF*_10_ > 100, and acceptable, *p*_*bonf*_ < .001, *BF*_10_ > 100, than the other two buildings. Furthermore, post hoc tests revealed that the Fiber Pavilion was rated as significantly more acceptable, *p*_*bonf*_ = .015, *BF*_10_ > 100, but not as more sustainable, *p*_*bonf*_ = 1, *BF*_01_ = 11.23, than the auditorium.

### 3.2. Biomimetic Hypothesis

The rmANOVAs revealed that neither perceived sustainability nor acceptability ratings differed for participants who were informed on the buildings’ bio-inspiration and those who were not, *F*(1, 236) = 0.05, *p* = .824, $\eta_{p}^{2}$ < .001 for perceived sustainability, *F*(1, 236) = 1.95, *p* = .163, $\eta_{p}^{2}$ = .01 for acceptability ratings. Thus, both parts of the hypothesis could not be supported. In addition, Bayesian rmANOVAs revealed strong evidence in favor of the null hypothesis, that is, for assuming that ratings are similar for participants who were or were not informed of the buildings’ bio-inspiration, both *BF*_01_ > 100 for perceived sustainability and acceptability.

### 3.3. Interaction Hypothesis

The rmANOVAs revealed no significant interaction between the between-subject factor, whether information on the buildings’ bio-inspiration was given or not, and the repeated measures factor type of building, neither regarding the perceived sustainability, *F*(2, 236) = 0.43, *p* = .651, $\eta_{p}^{2}$ < .001, nor regarding the perceived acceptability, *F*(2, 236) = 0.90, *p* = .407, $\eta_{p}^{2}$ < .001. Thus, the interaction hypothesis could not be supported. Bayesian rmANOVAs furthermore revealed strong evidence in favor of the null hypothesis, that is, for excluding the interaction from the model for both the perceived sustainability, *BF*_*excl*_ = 21.11, and the acceptability, *BF*_*excl*_ = 16.01.

### 3.4. Mediation Hypothesis

As there was no evidence that the perceived acceptability or perceived sustainability differed depending on the information regarding bio-inspiration, we did not perform any further analyses on a mediation between these two effects.

### 3.5. Exploratory Analyses

#### 3.5.1. Follow-Up

rmANOVAS on the data from the follow-up questionnaire mirror the result pattern regarding acceptability and perceived sustainability. There was no significant difference at the follow-up between participants who were or were not informed on the buildings’ bio-inspiration , neither regarding the perceived acceptability of the buildings, *F*(1, 206) = 1.95, *p* = .243, $\eta_{p}^{2}$ = .007, nor regarding the perceived sustainability of the buildings, *F*(1, 206) = 0.05, *p* = .252, $\eta_{p}^{2}$ = .006. Bayesian rmANOVAs revealed strong evidence in favor of the null hypotheses, i.e., that ratings do not differ for participants who were or were not informed on the buildings’ bio-inspiration, regarding both the perceived acceptability, *BF*_01_ > 100, and the perceived sustainability, *BF*_01_ > 100.

The rmANOVAs revealed that there was no significant interaction at the follow-up between the factor information on bio-inspiration and the factor type of building, neither regarding the perceived acceptability of the buildings, *F*(2, 206) = 0.16, *p* = .852, $\eta_{p}^{2}$ < .001, nor regarding the perceived sustainability of the buildings, *F*(2, 206) = 0.09, *p* = .914, $\eta_{p}^{2}$ < .001. Bayesian rmANOVAs furthermore revealed strong evidence in favor of excluding the interaction from the model, both for the perceived acceptability, *BF*_*excl*_ = 25.22, and the perceived sustainability, *BF*_*excl*_ = 28.49.

#### 3.5.2. Correlations Between Acceptability and Sustainability

Overall, there was a strong positive correlation between the buildings’ perceived acceptability and perceived sustainability, $\tau_{b}$ > .5, *p* < .001, *BF*_10_ > 100, for all three buildings.

### 3.6. Manipulation Checks

We coded whether or not the participants mentioned any inspiration from nature generally or from specific natural entities (like, e.g., specific plants, animals, or anatomic features). From those participants who received information about bio-inspiration, 92.5%, 88.43%, and 85.12% mentioned some inspiration from nature for the pavilion, the auditorium, and the shell, respectively, while of those participants who did not receive information about bio-inspiration, only 4.27%, 10.26%, and 12.82%, respectively, assumed that there might be some inspiration from nature.

### 3.7. Analyses of Socio-Political Attitudes

We checked for differences between participants who were or were not informed on the buildings’ bio-inspiration regarding their socio-political attitudes. We found no differences regarding scores on the cultural cognition worldview and regarding individual concerns, except for the item “worries concerning own health”. In addition, the groups did not differ regarding participants’ preferred political parties (for further details, see Appendix E and Appendix A on OSF).

## 4. Discussion

### 4.1. Discussion of the Current Study Results

In the current study, we investigated whether or not laypersons indicate a biomimetic bias. We derived the research question of whether or not there is a biomimetic bias in analogy to two different lines of research: first, in analogy to the biomimetic promise, and second in analogy to the natural-is-better bias. In detail, we examined whether or not laypersons perceive technical solutions—in the case of our study buildings—as more sustainable and more acceptable when they are framed as biomimetic buildings. To this end, participants assessed three different buildings in terms of their perceived sustainability and acceptability, with half of the participants being informed that these buildings were biomimetic and the other half of the participants not receiving this information.

The results do not differ in perceived sustainability or acceptability between participants who were informed about the buildings’ biomimetic nature and those who were not, indicating that there is no biomimetic bias. The Bayesian analyses indicate that our results are not merely absence of evidence but also *evidence of absence*—thus, our findings favor the hypothesis that there is no biomimetic bias. Furthermore, our findings provide evidence that there is no interaction (regarding the perceived sustainability or acceptability) between the type of building and whether participants did or did not receive information on the buildings’ biomimetic nature. Additionally, the assessments regarding perceived sustainability and acceptability were stable in the follow-up, which started one week after the main experiment. Thus, we conclude that the framing of buildings as biomimetic does not lead to a bias when assessing perceived sustainability and acceptability.

In line with Huijts et al. [36], Milchram et al. [37], Fenn et al. [38], Al-Emran [39], the measurements of perceived sustainability and acceptability correlated strongly. Participants who rated the buildings as more sustainable also rated their acceptability as higher. Such a correlation demonstrates that it is worthwhile to strive for sustainable, novel technical solutions to increase acceptability for the respective innovation.

The manipulation checks confirm that most participants who were not informed about the buildings’ biomimetic nature did not assume they were biomimetic, while those who received the information correctly recalled this information. Therefore, the similar perceived sustainability and acceptability ratings between the groups cannot be attributed to unclear condition separation. Consequently, we did not proceed with testing further hypotheses based on the assumption of a biomimetic bias. However, we found that the different buildings were perceived differently in terms of sustainability and acceptability, suggesting that participants were not generally indifferent to the information provided on each building.

The absence of a biomimetic bias suggests that “laypersons” evaluate the perceived sustainability and acceptability of biomimetic buildings based on factual information rather than on an inherent preference for biomimetic or bioinspired designs. This contrasts with the well-documented natural-is-better bias often observed in consumer goods, healthcare, and food contexts [6,26]. Thus, the current results underscore the importance of studying potential biases for each respective domain.

In addition, our study emphasizes the importance of transparency and detailed communication when promoting biomimetic or bioinspired technical solutions. Simply labeling a building as “inspired by nature” is insufficient to influence public perceptions of its sustainability or acceptability. Instead, providing detailed information regarding the ecological, economic, and social benefits of technical solutions is crucial to garnering public support. At least in the context of our study, this crucial information could be presented to laypersons in a relatively brief manner and still be well understandable. Thus, the way we presented information on the bioinspired buildings to participants in our study could potentially serve as a kind of showcase example for practitioners (e.g., in science communication) on how laypersons can be informed on biomimetic technical solutions in everyday contexts.

### 4.2. Future Research

Our results underscore the importance of carrying out research that is specific to domain and context when studying potential biases in the perception of technology caused by the technologies’ “naturalness”. Finding robust evidence for biases caused by natural ingredients to products that are ingested, or in other forms directly applied to the human body, is not sufficient for assuming that there are biases for other products and/or other forms of “naturalness”.

The fact that participants evaluated the buildings’ sustainability and acceptability independently from their biomimetic nature might be taken as a hint that laypersons’ conceptualizations of sustainability differ from the conceptualization of experts, who assign a normative value in terms of sustainability to biomimetics. In other words, while experts might attribute inherent sustainability to biomimetic or bioinspired designs due to their alignment with natural principles, laypersons do not connect biomimetic buildings with sustainability and instead base their judgments on tangible, presented benefits. This calls for future research into how sustainability is understood by the public and how expert assumptions align with these perceptions, because assuming that motifs that are prominent in experts’ discourse are shared by laypersons can be misleading. Specifically regarding biomimetic buildings, it might be interesting to conduct research on which building features, such as the use of a natural building material like wood or plant fibers, or which types of design language, such as biomorphic shapes, drive the differences in perceived sustainability between buildings.

Despite its strengths, our study has some specific limitations that warrant further investigation. The participant sample was limited to German-speaking individuals, which may restrict the generalizability of the findings to other cultural, linguistic, or geographical contexts. Future studies should explore whether or not the absence of a biomimetic bias holds across diverse cultural settings and in different types of biomimetic technologies beyond buildings. Additionally, while the study focused on sustainability and acceptability, other factors, such as aesthetic appeal, cost, and usability, could influence public perceptions of biomimetic buildings. Expanding the scope of inquiry to include these dimensions would provide a more comprehensive understanding of public attitudes. Future studies should also investigate the impact of various stakeholders—such as private entities, the state, or public institutions—on the allocation of costs associated with biomimetic solutions, as this could significantly shape both their implementation and public acceptance.

### 4.3. Practical Recommendations

For the successful adoption of biomimetic buildings, stakeholders should focus on clearly demonstrating their sustainability credentials rather than relying on their biomimetic nature as a persuasive tool on its own. Educational campaigns and public communication strategies should prioritize factual evidence of benefits, addressing ecological, economic, and social dimensions explicitly.

## 5. Conclusions

Our study challenges the assumption of a universal biomimetic bias in public perceptions of technologies’ sustainability and acceptability. The findings highlight the importance of evidence-based communication on specific benefits and suggest that the successful promotion of biomimetic or bioinspired technologies requires a more nuanced understanding of public conceptualizations of sustainability. By addressing this, future research can further bridge the divide between expert knowledge and layperson perceptions, paving the way for a shared discourse on and the broader adoption of sustainable innovations. Our results also support the hypothesis that there is no “biomimetic design language” that enables laypersons to judge biomimetic buildings or other products as inspired by living nature based on the features of these buildings/products.

## Figures and Tables

**Figure 1 biomimetics-10-00086-f001:**
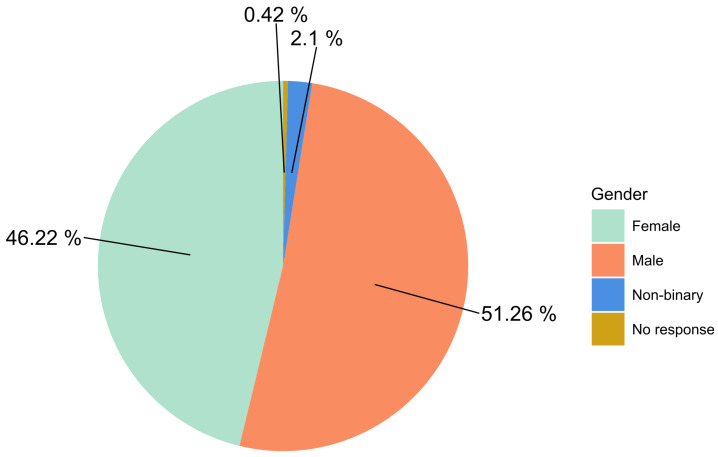
Gender distribution in percent of the final sample of participants (*n* = 238).

**Figure 2 biomimetics-10-00086-f002:**
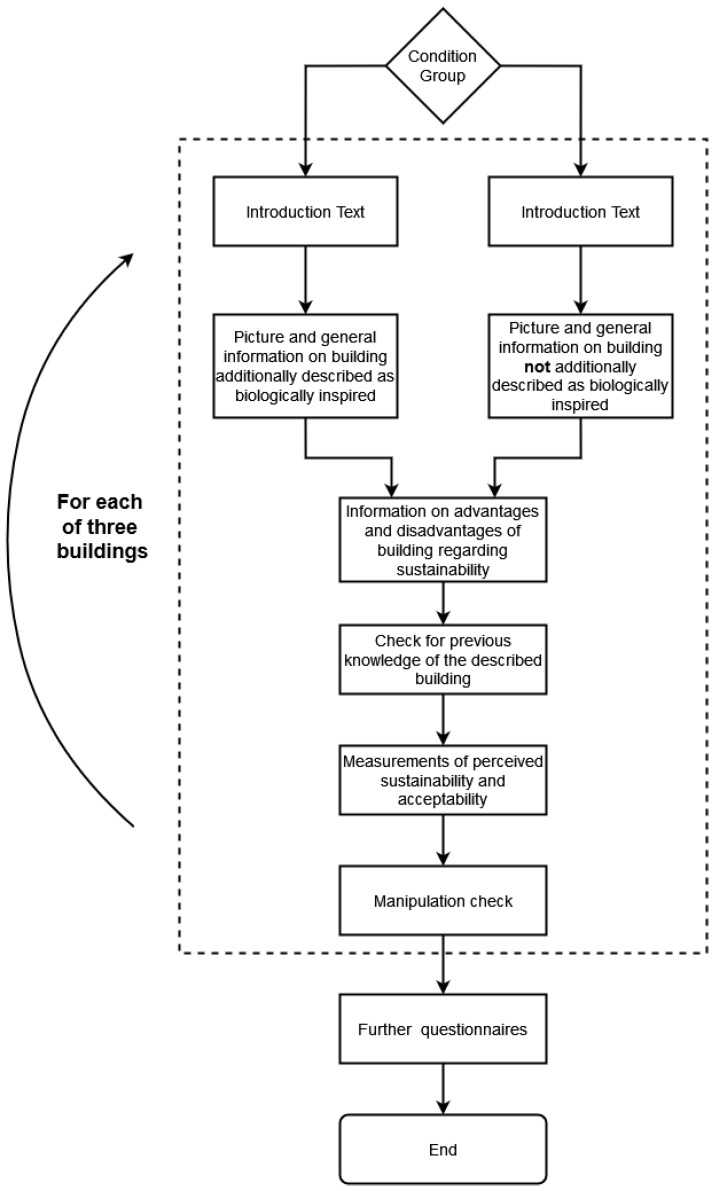
Flowchart of the experimental procedure of the main study.

**Figure 3 biomimetics-10-00086-f003:**
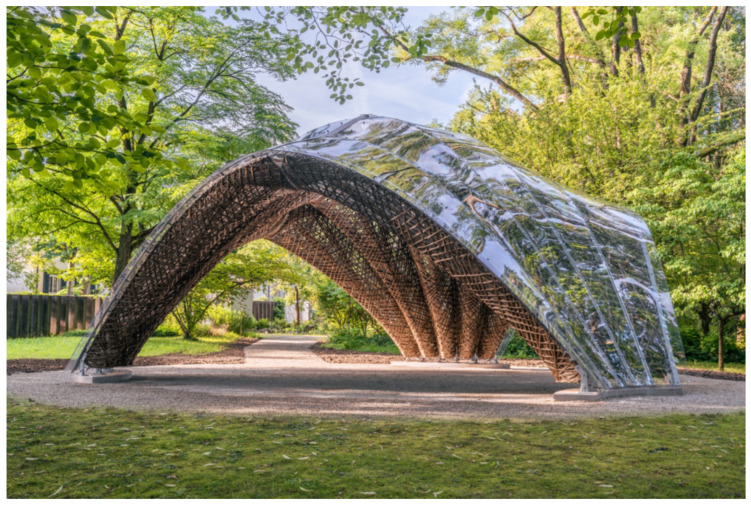
Fiber Pavilion. (Source: ©ICD/ITKE/IntCDC Universität Stuttgart, Rob Faulkner).

**Figure 4 biomimetics-10-00086-f004:**
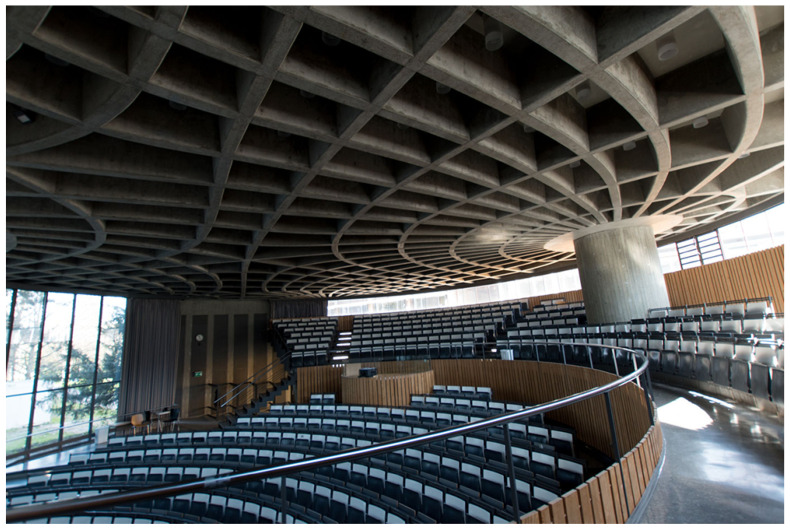
Auditorium. (Source: Universität Freiburg, https://www.osa.uni-freiburg.de/campustour/institutsviertel/hl, accessed on 23 November 2024).

**Figure 5 biomimetics-10-00086-f005:**
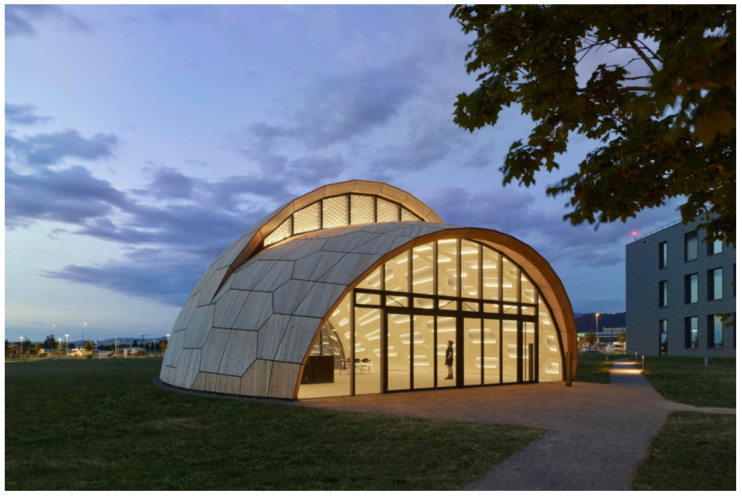
Wooden Shell. (Source: ©ICD/ITKE/IntCDC Universität Stuttgart, Foto: Roland Halbe).

**Figure 6 biomimetics-10-00086-f006:**
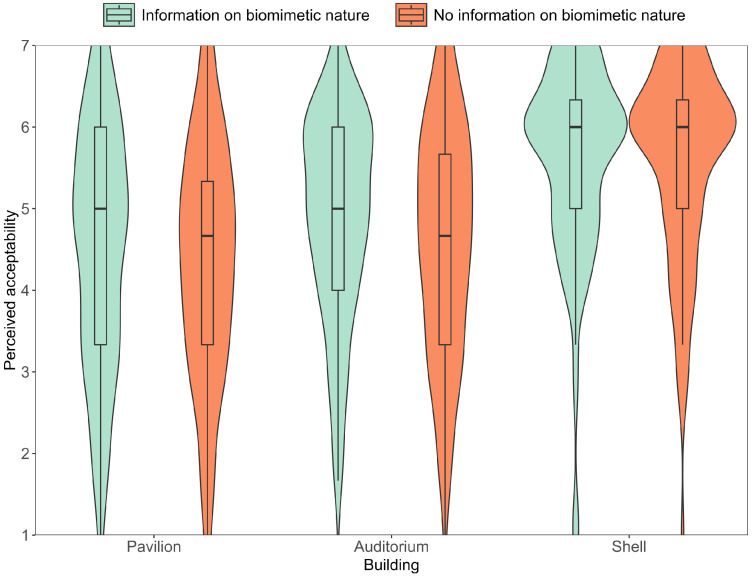
Average perceived acceptability. The boxplot shows median and interquartile range. Whiskers indicate the 1.5 interquartile range.

**Figure 7 biomimetics-10-00086-f007:**
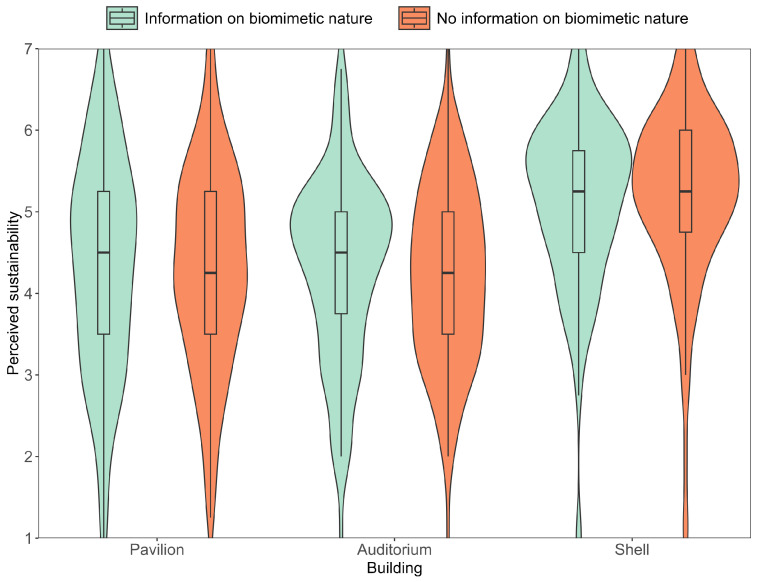
Average perceived sustainability. The boxplot shows median and interquartile range. Whiskers indicate the 1.5 interquartile range.

## Data Availability

Data can be downloaded together with data analyses, materials, and preregistration at https://osf.io/uhzgj/.

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
