# Peer review of "Challenging the Biomimetic Promise—Do Laypersons Perceive Biomimetic Buildings as More Sustainable and More Acceptable?"

_biomimetics, 2025, doi:10.3390/biomimetics10020086_

Round 1

Reviewer 1 Report

Comments and Suggestions for Authors

This article is accepted for publication as it is. 

Author Response

Dear reviewer,
We sincerely thank you for your positive evaluation of our manuscript. We are delighted to hear that you found our work suitable for publication and appreciate your recommendation to proceed with the article as it is.

Should you have any additional comments or suggestions in the future, we would be more than happy to address them. Thank you once again for your support and for endorsing our work for publication.

Best regards,
Michael Gorki (on behalf of the authors)

Reviewer 2 Report

Comments and Suggestions for Authors

Dear Authors,

I would like to express my sincere gratitude for your hard work on this paper.

The paper explores whether labeling buildings as biomimetic influences public perceptions of sustainability and acceptability. The authors tested three biomimetic buildings at the University of Freiburg. The results showed no significant impact of the biomimetic label but revealed differences in perceptions based on building design. Lastly, the authors emphasize the need for clear communication about actual sustainability benefits to gain public acceptance.

The topic and the overall approach are highly relevant, and the obtained results are compelling. Labeling alone does not necessarily enhance perceptions, and in a world plagued by greenwashing, this type of research addresses critical issues, such as public awareness and understanding of these pressing matters.

The manuscript is clear, well-structured in a classical format, easy to navigate, and relevant to the field. I would even describe it as captivating in many ways. The study is scientifically sound, with proper statistical analysis, and well-designed to test the hypothesis. However, I find it a bit too long and believe it would benefit from some reduction or the replacement of text with additional figures (see further comment). Nonetheless, congratulations to the authors on an excellent contribution!

Despite this, I have some minor comments:

-         Please reformulate the section on the “mediation hypothesis description” (lines 178–180), as it is currently unclear and difficult to understand.

-          The figures are appropriate and well-referenced. However, instead of a long text describing participants' demographics (Section 2.2: Participants), it would be beneficial to include graphical information, as it would make the data easier to interpret, especially given the density of the paper (which is a good thing). The use of violin plots with embedded box plots is an excellent and clever choice for visualizing results—nothing to say here.

-          The discussion and conclusion sections are consistent and coherent with the findings. The authors acknowledge their study's limitations and draw a future research path. It would be interesting to conduct a similar study in other geographical contexts or, as the authors say, under “different cultural cognition worldviews,” as the results might differ (a potential bias in the current study). Nonetheless, I believe this research is a strong scientific contribution.

-          The text is well referenced. However, I noticed that 18 of the 57 references (32%) related to the fields of “Biomimetics” and “Sustainability” are older than five years. This may indicate that some supporting information is outdated, especially given the significant advancements in these fields over the past five years. My suggestion here: support your arguments with recent work, or please argue the selection of these references.

-          I found five self-citations, all related to the subject and mostly within the last five years. Considering the number of authors involved, this is perfectly acceptable.

-          Ethical considerations were appropriately disclosed, and I do not have any ethical concerns regarding this work.

Final remark: My only regret is that this study and paper are not mine. Well done!

Author Response

Dear reviewer,

We sincerely thank you for your thoughtful and positive evaluation of our manuscript. We greatly appreciate your constructive suggestions, which have helped us refine and strengthen our work.

In response to your recommendations, we have carefully revised the manuscript to address the points you raised. We believe these revisions have enhanced the clarity and depth of the paper. Please find a detailed point-by-point response to your comments below. We have marked all changes in our article (that go beyond an additional proofreading) in green color.

Thank you once again for your valuable feedback and for contributing to the improvement of our manuscript.

Best regards,
Michael Gorki (on behalf of the authors)

Comment 1: Please reformulate the section on the “mediation hypothesis description” (lines 178–180), as it is currently unclear and difficult to understand.

Response 1: We agree that the description of the mediation hypothesis could be improved. We have revised this part to improve clarity and make it easier to understand. See p. 5 (lines 181-183).

Comment 2: The figures are appropriate and well-referenced. However, instead of a long text describing participants' demographics (Section 2.2: Participants), it would be beneficial to include graphical information, as it would make the data easier to interpret, especially given the density of the paper (which is a good thing). The use of violin plots with embedded box plots is an excellent and clever choice for visualizing results—nothing to say here.

Response 2: We followed your suggestion to present participants’ demographic data visually. Specifically, we created a pie chart for gender distribution, replacing the lengthy parenthetical description. Additionally, responding to another reviewer’s comment, we provided more detailed demographic data in two tables added to the Appendix (Appendix B). This ensures that the text in Section 2.2 is not further extended while presenting all relevant information concisely.

Comment 3: The discussion and conclusion sections are consistent and coherent with the findings. The authors acknowledge their study's limitations and draw a future research path. It would be interesting to conduct a similar study in other geographical contexts or, as the authors say, under “different cultural cognition worldviews,” as the results might differ (a potential bias in the current study). Nonetheless, I believe this research is a strong scientific contribution.

Response 3: We fully agree with your comment on the importance of conducting similar studies in other geographical contexts. We explicitly added the geographical aspect you mentioned in this context in the discussion, see p. 16.

Comment 4: The text is well referenced. However, I noticed that 18 of the 57 references (32%) related to the fields of “Biomimetics” and “Sustainability” are older than five years. This may indicate that some supporting information is outdated, especially given the significant advancements in these fields over the past five years. My suggestion here: support your arguments with recent work, or please argue the selection of these references.

Response 4: Generally, we have cited the most relevant papers in the field of our research. It is true that some of the cited papers are older than 5 years, but we believe that they are still relevant – both for the field and for our article. Following your suggestion, we made again a literature search to make sure that we do not miss to point to recent developments or debates. Based on the literature search, we included two additional references to recent work. Specifically, we included Broeckhoven & Winters (2023) on the danger of the naturalistic fallacy in the biomimetic promise and Oguntona & Aigbavboa (2023) on sustainability benefits of biomimetic solutions in built environments. Apart from these additions, we decided to keep the references as they are. On a more general level, your comment proves the relevance of our research, as there has obviously been too little research done on this important topic in recent years.

Reviewer 3 Report

Comments and Suggestions for Authors

- It is suggested to better clarify the main objective of the study in the introduction. For instance, how does this study contribute to improving the understanding of the relationship between biomimetic design and sustainability?

- While the methodology is explained reasonably well, further clarification is needed regarding participant selection. Was cultural or demographic diversity considered?

- The method for addressing potential biases that might arise from presenting different information to the two groups is not mentioned. This should be clarified to avoid doubts about the results.

- A deeper analysis of the reason for the absence of a "biomimetic bias" is recommended. It might be beneficial to include an explanation of other factors that influenced participants' perceptions, such as building cost or aesthetics.

It is suggested to expand the conclusion to include practical examples of how to improve public communication regarding the benefits of biomimetic design.

Comments on the Quality of English Language

Some phrasing needs improvement for better clarity, particularly in the introduction and the section on the study's significance.

Author Response

Dear reviewer,

Thank you for your thorough and thoughtful review of our manuscript. We appreciate your constructive criticism and the opportunity to address the points you raised, which have been valuable in improving the quality and clarity of our work.

While we are encouraged by your overall positive assessment, we have carefully considered and addressed the more critical aspects of your feedback. We believe these revisions have strengthened the manuscript and ensured that it more effectively communicates our research contributions. Please find a detailed point-by-point response to your comments below, which outlines the specific changes we have made in response to your feedback. We have marked all changes in our article (that go beyond an additional proofreading) in green color.

Thank you once again for your valuable insights and for helping us refine our manuscript.

Best regards,
Michael Gorki (on behalf of the authors)

Comment 1: It is suggested to better clarify the main objective of the study in the introduction. For instance, how does this study contribute to improving the understanding of the relationship between biomimetic design and sustainability?

Response 1: In order to add information on how our results might inform communication to the public regarding biomimetic design and perception of sustainability, we now foreshadow in the introduction that we will elaborate on this topic in the discussion section (see point 4 of this review). We conjecture that our study serves as an example for public communication (see p. 15 as explained below).

Comment 2: While the methodology is explained reasonably well, further clarification is needed regarding participant selection. Was cultural or demographic diversity considered?

Response 2: We have expanded Section 2.2 to provide more detail on participant selection. Additionally, we now include descriptive statistics in the appendix on the diversity of our sample in terms of ethnicity and net household income (Appendix B. Additional Demographic Characteristics of the Sample). Reference to this appendix as well as to the appendices on socio-political attitudes of our participants are now explicitly included in Section 2.2, when discussing the diversity of our sample.

Comment 3: The method for addressing potential biases that might arise from presenting different information to the two groups is not mentioned. This should be clarified to avoid doubts about the results.

Response 3: Thank you for highlighting this issue. We have revised Section 2.3.1 to state more explicitly that the only difference between the two groups was the inclusion of additional information on the biomimetic aspect of the buildings for one group (i.e., our experimental manipulation).

Comment 4: It is suggested to expand the conclusion to include practical examples of how to improve public communication regarding the benefits of biomimetic design

Response 4: We agree that practical examples of how to improve public communication regarding benefits of biomimetic design can be very helpful especially for practitioners (e.g., in science communication). Based on the results from both our pretest as well as our main study we think that the way we informed participants in our study can in itself serve as an example of how public communication can be detailed, well understandable and at the same time brief enough for the application in everyday contexts. Since we directly include all textual and visual materials used to inform participants in the study in the methods section, they are readily accessible to readers. We now emphasize the aspect that information presented to participants in our study can serve as an example for public communication in our discussion (p. 15).

Comment 5: A deeper analysis of the reason for the absence of a "biomimetic bias" is recommended. It might be beneficial to include an explanation of other factors that influenced participants' perceptions, such as building cost or aesthetics.

Response 5: We absolutely agree that deeper analysis of the reasons for the absence of a biomimetic bias would be highly interesting. That said, we acknowledge that future studies should take additional factors like aesthetics and cost into account. We now further elaborate on this in the Discussion section (4.2. Future Research). In addition to mentioning a potential impact of aesthetics and building costs, we now add that the impact of various stakeholders (private entities, the state, or public institutions) on the allocation of cost as a potential factor that should be addressed in future studies.

Comment 6: Some phrasing needs improvement for better clarity, particularly in the introduction and the section on the study's significance.

Response 6: We conducted additional proofreading and made several linguistic and stylistic improvements, particularly in the introduction.

Round 2

Reviewer 3 Report

Comments and Suggestions for Authors

After carefully reviewing the revised manuscript, I can confirm that the necessary corrections have been made comprehensively and satisfactorily.

In my opinion, the manuscript is now ready for publication in the journal